# Co-Modality Graph Contrastive Learning for Imbalanced Node Classification

**Yiyue Qian**[1], **Chunhui Zhang**[2], **Yiming Zhang**[3], **Qianlong Wen**[1],
**Yanfang Ye**[1*], **Chuxu Zhang**[2*]

yqian5@nd.edu, chunhuizhang@brandeis.edu, yxz2092@case.edu,
qwen@nd.edu, yye7@nd.edu, chuxuzhang@brandeis.edu

[1]Department of Compute Science and Engineering, University of Notre Dame, USA
[2]Department of Computer Science, Brandeis University, USA
[3]Department of Computer and Data Sciences, Case Western Reserve University, USA

## Abstract

Graph contrastive learning (GCL), leveraging graph augmentations to convert graphs into different views and further train graph neural networks (GNNs), has achieved considerable success on graph benchmark datasets. Yet, there are still some gaps in directly applying existing GCL methods to real-world data. First, handcrafted graph augmentations require trials and errors, but still can not yield consistent performance on multiple tasks. Second, most real-world graph data present class-imbalanced distribution but existing GCL methods are not immune to data imbalance. Therefore, this work proposes to explicitly tackle these challenges, via a principled framework called *Co-Modality Graph Contrastive Learning* (**CM-GCL**) to automatically generate contrastive pairs and further learn balanced representation over unlabeled data. Specifically, we design inter-modality GCL to automatically generate contrastive pairs (e.g., node-text) based on rich node content. Inspired by the fact that minority samples can be "forgotten" by pruning deep neural networks, we naturally extend network pruning to our GCL framework for mining minority nodes. Based on this, we co-train two pruned encoders (e.g., GNN and text encoder) in different modalities by pushing the corresponding node-text pairs together and the irrelevant node-text pairs away. Meanwhile, we propose intra-modality GCL by co-training non-pruned GNN and pruned GNN, to ensure node embeddings with similar attribute features stay closed. Last, we fine-tune the GNN encoder on downstream class-imbalanced node classification tasks. Extensive experiments demonstrate that our model significantly outperforms state-of-the-art baseline models and learns more balanced representations on real-world graphs. Our source code is available at https://github.com/graphprojects/CM-GCL.

## 1 Introduction

Contrastive learning, aiming to contrast instance pairs generated from unlabeled data to train better representation models, has attracted considerable attention. Inspired by the consistent success of contrastive learning in computer vision and natural language processing [3, 2, 26], an increasing number of works have started to investigate the great potential of contrastive learning on graphs [45, 52, 38]. Most existing graph contrastive learning (GCL) models leverage different graph transformation methods to augment graph data into different views, and further train graph neural networks (GNNs)

---

*Corresponding authors.

by maximizing the similarity between positive pairs while minimizing the similarity between negative pairs. In particular, PyGCL [52] summarizes nine categories of graph augmentations from different levels (e.g., node dropping, edge perturbation, and graph diffusion), and implements different augmentation combinations to transform graph into different views for contrastive learning.

Yet, directly applying existing GCL models [28, 27, 31, 45] from the controlled benchmark graph datasets to uncontrolled real-word graph datasets still has some gaps. First, notwithstanding suitable handcrafted augmentation methods need professional knowledge as well as trials and errors, the consistent excellent performance on multiple tasks can not be guaranteed [52]. Second, node classes in most real-world graphs present imbalanced distribution (e.g., Zipf long-tail distribution). For instance, the amount of illicit activities (e.g., malicious code/repository [44, 10, 7, 43], illicit drug/crimeware trading [30, 48]) on the Internet and social networks is much less than that of normal activities [33]. To tackle these challenges, we aim to design a novel GCL model that automatically generates contrastive pairs without too much effort, but also can generalize well in real-world graphs with the class imbalance problem.

Accordingly, inspired by existing cross-modality contrastive learning methods in computer vision and NLP [32, 53], we naturally hypothesize that node embeddings in graph and the corresponding content embeddings (e.g., text embedding) in different modalities will be closer if we train GNN encoder and the content encoder (e.g., text encoder) simultaneously during contrastive learning. In this way, we can automatically generate contrastive pairs, e.g., (node-text), based on the natural node content, which is called inter-modality GCL. However, recent studies [14, 42] demonstrate that contrastive learning can alleviate the data imbalance problem but fails to be fully immune to the data imbalance issue. Therefore, motivated by the fact that minority data may be easily "forgotten" by pruning deep neural networks [13], we extend the ad-hoc compression tool, network pruning, to GCL framework and further discover minority samples not yet well represented by encoders during co-modality GCL.

To summarize, we propose a novel framework called *co-modality graph contrastive learning* (**CM-GCL**) to automatically generate contrastive pairs from unlabeled data and further learn more balanced node representations. As illustrated in Figure 1, we first prune the graph encoder and content encoder in inter-modality GCL to uncover more minority samples, and further co-train two encoders via optimizing the inter-modality contrastive loss. Besides, we propose to ensure that node embeddings with similar attribute features should also stay closed. Therefore, we propose intra-modality GCL that generates contrastive pairs based on the similarities among node attribute features and further co-trains non-pruned graph encoder and pruned graph encoder. By the aforementioned steps, CM-GCL can boost minority nodes weights in contrastive loss and lead to implicit loss re-balancing. Finally, we fine-tune the pre-trained GNN encoder on the class-imbalanced node classification tasks. To conclude, our work makes the following contributions:

- As handcrafted graph augmentations are not efficient on real-world graph datasets, we devise a co-modality framework to automatically generate contrastive pairs. To the best of our knowledge, this is the first work on GCL that co-trains encoders in different modalities to facilitate node representation learning.

- To address the class imbalance issue on graphs, our model extends network pruning to prune encoders in co-modality GCL for discovering minority samples and further learning more balanced node representations.

- Extensive experiments on multiple real-world graph datasets show the effectiveness of our model by comparing CM-GCL with state-of-the-art methods.

## 2   Related Work

**Graph Contrastive Learning.** Most GCL models [31, 45, 38, 46, 40, 39] leverage different types of data transformations to augment graph into different views and further train an encoder by discriminating positive pairs and negative pairs generated from unlabeled data. Specifically, GraphCL [45] designs four types of graph augmentations to generate contrastive pairs in different views for contrastive learning. GCC [31] implements the random walk for each node to sample subgraphs as augmentation. In the heterogeneous domain, HeCo [38] employs cross-view (schema view and meta-path view) contrastive learning to train an encoder. These handcrafted data augmentations require trials and errors, and sufficient domain knowledge, but still cannot yield consistent excellent performance on multiple tasks [45]. Hence, we naturally propose to automatically generate contrastive

pairs for graph data. Inspired by CLIP [32], a cross-modality contrastive learning model on image and text data, we devise cross-modality GCL that leverages the rich node content to automatically generate useful contrastive pairs without resorting to handcrafting or expert knowledge.

**Data Imbalance in Self-supervised Learning.** Data imbalance is very common in real-world applications (e.g., anomaly detection [5, 6]). Most existing methods against data imbalance focus on supervised learning (e.g., re-sampling [1, 24], re-weighting [16, 35], and GraphSMOTE [51]). Recent works [42, 15, 14] start to explore the benefits of the balanced feature space from self-supervised learning. Yang and Xu [42] first studied the label bias in self-supervised learning and concluded that classifiers that are pre-trained in a self-supervised manner consistently outperform their corresponding supervised baselines. Kang et al. [15] found that contrastive learning can present a more balanced feature space over imbalanced datasets by comparison with supervised learning. Based on this, SDCLR [14] validates that contrastive learning is not fully immune to data imbalance. All mentioned works study the imbalance problem of image. Motivated by these works, we devise an algorithm that can well mitigate the data imbalance in our proposed co-modality GCL framework.

**Network Pruning.** Network pruning [19, 22, 20] has been considered as a popular compression tool for DNN models. In recent years, some works [8, 13, 14] have started to explore the deeper connection with DNN memorization and generalization. Specifically, Frankle and Carbin [8] found that pruning a dense DNN can identify "significant subnetwork" whose testing accuracy is comparable to the one in the original network. Hooker et al. [13] thought a small data subset, i.e., data belonging to ambiguous classes and minority classes, is more likely to be impacted by the introduction of sparsity via network pruning. Based on the conclusion, Jiang et al. [14] constructed a self-competitor via network pruning to study the imbalanced image in contrastive learning. In this work, we introduce network pruning in co-modality contrastive learning to uncover minority nodes and further gain the more balanced representation.

## 3 Preliminary

**Graph Neural Networks.** Given a graph $G = (\mathcal{V}, \mathcal{E}, \mathcal{X})$, where $\mathcal{V}$ is the set of different types of nodes, $\mathcal{E} \subseteq \mathcal{V} \times \mathcal{V}$ is the set of edges, and $\mathcal{X}$ is the attribute feature set. We aim to learn the node representation by considering both structure and node attribute features. Current GNNs (e.g., GCN [18], GAT [37], GraphSAGE [9], and HetGNN [47]) follow a messaging-passing framework and has gained consistent performance on various graph data. In this paper, we choose GCN as the graph encoder to learn the node embedding $h_G^i \in \mathbb{R}^{d_G}$ corresponding to node $v_i$. In particular, GCN is formulated as $H^{l+1} = \sigma(\widetilde{A} H^l W^l)$, where $H^{l+1}$ denotes the node representations at $l+1$ layer, $\widetilde{A}$ is a symmetric normalization of $A$ with self-loop, $W^l$ is the weight matrix at $l$-th layer, and $\sigma$ is the activation function. We use $f_G$ to denote the graph encoder and $h_G$ to denote the node embedding. Besides GCN, we also discuss GAT and GraphSAGE as the graph encoder in experiments.

**Text Pre-Trained Model.** In most real-world graphs, nodes contain rich text information. Most existing graph representation learning methods [18, 9, 50, 49, 41] extract the keyword from text and convert the rich text content into sparse bag-of-words feature vector, and further utilize the extracted features to facilitate the graph representation learning. Instead of directly utilizing the bag-of-words feature vectors as the attribute features for nodes, we propose to fine-tune the pre-trained language model on the handy unlabeled dataset within the graph. In this work, we leverage the pre-trained transformer language model (e.g., DistilBERT [34], a distilled version of BERT) as the text encoder and further fine-tune the text encoder with co-modality contrastive loss. The structure of DistilBERT we implemented is a base size with a 6-layer 512-wide model along with 8 attention heads. Text sequence is bracketed with [SOS] and [EOS] tokens and we take the activation of the highest layer of transformer at [EOS] token as text representation $h_T \in \mathbb{R}^{d_T}$. In this paper, $f_T$ denotes the text encoder. After fine-tuning $f_T$ via optimizing co-modality contrastive loss, we replace the bag-of-words feature vectors with fine-tuned attributes for all nodes. Besides DistilBERT, we also discuss other transformer models (i.e., BERT) as the text encoder and more details can be found in Section 5.5.

**Image Pre-Trained Model.** Except for text information, image content is also very common for nodes in graph (e.g., Instagram). We implement two different architectures (i.e., ResNets and Transformer) as the image encoder. As ResNets is widely used and has been proven with excellent performance, we implement ResNet50 [11] as the base architecture. Following [12], we make similar

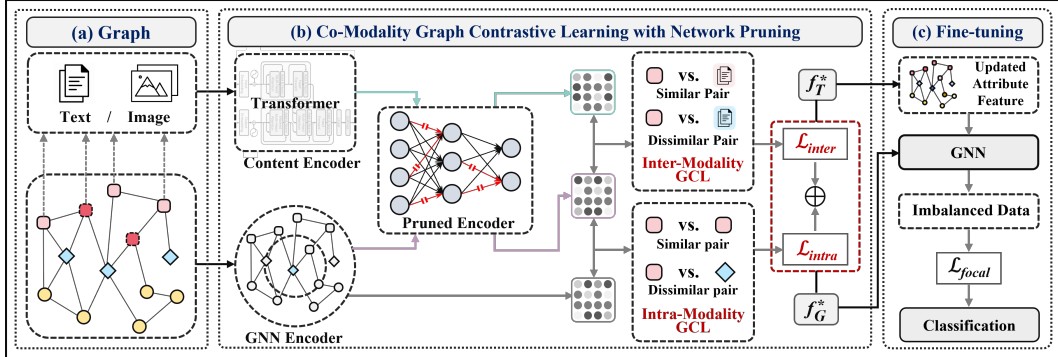

Figure 1: The overall framework of CM-GCL: (a) It constructs graph to depict content data (e.g., image and text) and relation data in networks; (b) It co-trains the pruned GNN and pruned content encoder in inter-modality GCL to ensure node embeddings and content embeddings referring to the same entity stay together; Intra-modality GCL that co-trains non-pruned GNN and pruned GNN is further proposed to ensure node embeddings with similar attributes stay close. (c) It updates attribute features via $f_T^*$ and the pre-trained GNN $f_G^*$ is fine-tuned on downstream class imbalance tasks.

modifications on ResNet and replace the global average pooling layer with an attention pooling mechanism. Besides, we consider the recently introduced Swin Transformer [23] as the image encoder and closely follow their implementations. We utilize $f_I$ to denote the image encoder to obtain the image representation $h_I \in \mathbb{R}^{d_I}$. After fine-tuning $f_I$ via contrastive loss optimization, we attach the fine-tuned image embedding to the corresponding nodes as the attribute features for downstream tasks.

**Problem 1.** *Co-Modality Graph Contrastive Learning for Imbalanced Node Classification. Given a graph $G = (\mathcal{V}, \mathcal{E}, \mathcal{X})$ built on multi-modality data along with imbalanced class label $Y$, we aim to build a GCL model over $G$ that can be further fine-tuned or applied to the downstream tasks with imbalanced labels (e.g., imbalanced node classification).*

## 4    Methodology

In this section, we present the details of CM-GCL (Figure 1) which includes three steps: (i) graph construction (Figure 1.(a)); (ii) co-modality graph contrastive learning with network pruning (Figure 1.(b)); (iii) model fine-tuning on downstream imbalanced node classification tasks (Figure 1.(c)).

### 4.1    Graph Construction

Graph has been proven to be effective in modeling real-world networks (e.g., academic networks and social networks). Merely taking the relational structure information among nodes into consideration is deemed insufficient to learn the node representations. Thus, to model real-world graph data for certain tasks (e.g., domain prediction for AMiner academic network), we depict both the informative content and rich structure information within graph data. Following existing works [18, 9, 37], we first convert the text content into sparse bag-of-words feature vectors attached to the corresponding nodes, which is denoted as $x_G$. For some graphs that have rich image content (e.g., Instagram), we implement the pre-trained image encoder (e.g., ResNet) to obtain the image feature vectors applied to the corresponding nodes. Besides that, different from existing works, we further fine-tune the content encoder (denoted as $f_T$ for text encoder or $f_I$ for image encoder) via co-modality contrastive learning. Inspired by CLIP [32] that co-trains the text encoder and the image encoder for the text image matching task, we view the rich content as the text (image) modality data and the corresponding graph structure with bag-of-words feature vectors as the graph modality data. We argue the corresponding data in different modalities can achieve agreements in the learned representation space.

### 4.2    Co-Modality Graph Contrastive Learning

After graph construction, we first propose inter-modality contrastive learning to automatically generate contrastive pairs. Then we introduce network pruning to discover minority nodes during contrastive learning. Last, we design intra-modality contrastive learning to ensure that node embeddings with similar semantics (attribute features) stay closed.

### 4.2.1 Inter-Modality Graph Contrastive Learning

The main idea of contrastive learning is to make representations maximize the agreement on sample pairs under proper graph augmentations. Yet, finding proper graph augmentation for every task is always resource-consuming [45]. Inspired by existing works [32, 53] in terms of cross-modality learning, we propose to fully utilize nodes' content to automatically generate contrastive pairs, and further co-train encoders in different modalities via contrastive learning. That is to say, we aim to learn two encoders such that embeddings in two modalities are close to each other in the learned space if they refer to the same entity, otherwise far away. In this paper, we mainly consider two combinations of modalities as illustrated in Figure 1.(a) (i.e., graph-text or graph-image). For simplicity, we discuss graph-text modalities as an example in this section, and will conduct experiments on graph-image modalities. Specifically, given a sample pair $(x_G^i, x_T^i)$, where $x_G^i$ is the original attribute feature (i.e., bag-of-words feature vector) of node $v_i \in \mathcal{V}$ and $x_T^i$ is the corresponding text content input, we feed $x_G^i$ to the graph encoder $f_G$ and $x_T^i$ to the text encoder $f_T$ respectively, and further obtain the corresponding representations $(h_G^i, h_T^i)$. Note that, as different types of nodes in $G$ may have different feature dimensions, we first transform the input attribute features of all nodes in $G$ to a common space. After obtaining the representation pair $(h_G^i, h_T^i)$, we apply nonlinear projection heads to convert embeddings from different modalities to the same space (denoted as $(z_G^i, z_T^i)$) for comparison, which can be formally defined as $z^i = \mathrm{MLP}(h^i)$. We apply separate projection heads for representations generated by the encoder in different modalities. Here we denote $(z_G^i, z_T^i)$ as the projected embedding pair. We consider $(z_G^i, z_T^i)$ as a similar pair (positive pair) if the node and the text content refer to the same entity. Otherwise, $(z_G^i, z_T^i)$ is viewed as a dissimilar pair (negative pair).

### 4.2.2 Network Pruning

We are aware that imbalanced data fails many supervised approaches built on balanced benchmarks. Whether contrastive learning is immune to data imbalance or not has remained controversial. Recent studies [14, 15, 42] argue and demonstrate that self-supervised contrastive learning can alleviate the data imbalance issue in representation learning but is not fully immune to data imbalance. Based on the above findings, we propose to alleviate the data imbalance issue during our proposed co-modality GCL. Some recent works [13, 14] discover that certain examples are particularly sensitive to sparsity via network pruning and these samples mostly impacted after pruning are termed as Pruning Identified Exemplars (PIEs) [13]. They further demonstrate that PIEs often show up in minority classes. Inspired by this, we extend network pruning to our proposed framework to uncover minority samples. Unlike pruning a single encoder for image model in the previous works [13, 14, 8], we co-prune multiple encoders in different modalities. With PIEs dynamically generated during contrastive training, more minority samples will be exposed along with training epochs, as we will show in Figure 2. In this paper, we employ the magnitude pruning method [8] that uses the absolute value of weights to rank their importance and removes weights that are below a certain threshold. Specifically, the detailed process is described as follows:

1. Randomly initialize encoders $f_G$ and $f_T$ with parameters $\theta_G^0$ and $\theta_T^0$ respectively.
2. At each iteration $l$, prune $e\%$ of smallest-magnitude parameters in $\theta_G^l$ and $\theta_T^l$ by creating the corresponding masks $m_G^l$ and $m_T^l$.
3. Apply the masks $m_G^l$ and $m_T^l$ to the feed-forward encoders $f_G$ and $f_T$ respectively, and calculate the corresponding contrastive loss.
4. Release the masks $m_G^l$ and $m_T^l$ on the masked parameters and update encoders' parameters via optimizing the contrastive loss.

In the above steps, the contrastive loss in inter-modality GCL is formulated as:

$$\mathcal{L}_{\mathrm{inter}} = -\log \sum_{v_i \in \mathcal{V}} \frac{\exp\left[\mathrm{sim}\left(\widetilde{z}_G^i, \widetilde{z}_T^i\right)/\tau_{\mathrm{inter}}\right]}{\sum_{p=1}^{2n} \mathbb{1}_{[i \neq p]} \exp\left[\mathrm{sim}\left(\widetilde{z}_G^i, \widetilde{z}_T^p\right)/\tau_{\mathrm{inter}}\right]}, \tag{1}$$

where $(\widetilde{z}_G^i, \widetilde{z}_T^i)$ is the projected embedding pair generated by the pruned encoders $\widetilde{f}_G$ and $\widetilde{f}_T$, $\mathrm{sim}(\widetilde{z}_G^i, \widetilde{z}_I^i)$ is the cosine similarity between $\widetilde{z}_G^i$ and $\widetilde{z}_I^i$, $\tau_{\mathrm{inter}}$ is the temperature value, and $n$ is the size of mini-batch. Following the iterative pruning process that repeatedly trains, prunes and releases weights in each round, it effectively boosts minority samples' weights in $\mathcal{L}_{\mathrm{inter}}$ and leads to implicit loss re-balancing. In this way, minority samples are more likely to be uncovered and further force the encoders to learn more about these minority samples.

### 4.2.3   Intra-Modality Graph Contrastive Learning

Inter-modality graph contrastive learning ensures that embeddings from different modalities w.r.t. the same entity stay closed in the projected space but fails to ensure that embeddings with similar original features from the same modality stay closed. We argue that nodes with similar attribute features naturally have similar semantics and their embeddings should be closer than those with dissimilar attribute features. However, most existing GNNs are mainly dominated by the relationships among nodes in graph [29]. In light of this, we propose intra-modality graph contrastive learning to balance the importance of local neighbor relations and attribute features, and further ensure that node embeddings with similar attribute features will stay closer in the projected space. Note that, we mainly focus on graph modality in intra-modality graph contrastive learning. Specifically, we first define $\mathcal{S}$ as the set of node pairs having the top-R similarity, which is formally defined as:

$$\mathcal{S} = \{(v_i, v_p) \mid \mathrm{sim}(x_G^i, x_G^p) \text{ in top-R of } [\mathrm{sim}(x_G^i, x_G^p)]_{p=1}^{N}, \ \forall v_i \in \mathcal{V}\}, \tag{2}$$

where $\mathrm{sim}(x_G^i, x_G^p)$ measures the cosine similarity of node attribute features between node $v_i$ and $v_p$, $R$ is the number of top pairs chosen for each node, and $N$ is the number of nodes in $G$.

The main difference between intra-modality GCL and inter-modality GCL lies in that, intra-modality GCL employs the graph encoder and the pruned graph encoder for contrastive training, while inter-modality GCL implements the graph encoder and the text (image) encoder for co-modality contrastive training. Hence, we have a dense graph encoder $f_G$ and a sparse graph encoder $\widetilde{f}_G$ pruned by iterative magnitude pruning. Given a sample pair $(x_G^i, x_G^p)$, where $i$ can equal $p$, it will be encoded by $f_G$ and $\widetilde{f}_G$ respectively. The projected embeddings $(z_G^i, \widetilde{z}_G^p)$ is regarded as a positive pair if $(v_i, v_p)$ in $\mathcal{S}$, otherwise, it will be viewed as a negative pair. The intra-modality GCL loss is defined as follows:

$$\mathcal{L}_{\mathrm{intra}} = -\log \sum_{v_i \in \mathcal{V}} \frac{\sum_{(v_i, v_p) \in \mathcal{S}} \exp \left[ \mathrm{sim} \left( \widetilde{z}_G^i, \widetilde{z}_G^p \right) / \tau_{\mathrm{intra}} \right]}{\sum_{(v_i, v_q) \notin \mathcal{S}} \exp \left[ \mathrm{sim} \left( \widetilde{z}_G^i, \widetilde{z}_G^q \right) / \tau_{\mathrm{intra}} \right]}. \tag{3}$$

By this way, for atypical and minority nodes, our model can amplify the representation differences between the pruned and non-pruned graph encoders in intra level, and these nodes' weights will be implicitly increased in the intra-modality contrastive loss $\mathcal{L}_{\mathrm{intra}}$.

After performing inter-modality graph contrastive learning and intra-modality graph contrastive learning, the overall objective of co-modality graph contrastive learning can be formulated as:

$$\mathcal{L} = \lambda \mathcal{L}_{\mathrm{inter}} + (1 - \lambda) \mathcal{L}_{\mathrm{intra}}, \tag{4}$$

where $\lambda$ is the trade-off hyper-parameter for balancing two loss terms. Pseudo-code of CM-GCL is provided in Section A of the Appendix.

### 4.3   Fine-tuning

After sufficient training on the co-modality GCL, different from most existing methods that directly fine-tune the pre-trained encoder $f_G^*$ on downstream tasks [31, 45], we adopt a multi-step protocol to evaluate the classification performance. (i) Representation learning via CM-GCL: we pre-train encoders by optimizing the pre-training loss $\mathcal{L}$ in Eq. 4; (ii) Update the attribute feature: we leverage the learned content encoder $f_T^*$ to get the updated attribute features; (iii) Fine-tuning the GNN encoder: we adopt the pre-trained weights from $f_G^*$ as the initialized parameters for the fine-tuning encoder. Then we train the GNN encoder together with the classifier (i.e., MLP) over imbalanced nodes. To address the class imbalance problem, we introduce Focal Loss [21] that applies a modulating term to the cross-entropy loss to focus on hard misclassified nodes. This can be considered as a dynamically scaled cross entropy loss, where the scaling factor can down-weight the contribution of easily classified nodes automatically during model fine-tuning and rapidly focuses the model on hard nodes. In particular, the supervised multi-class focal loss $\mathcal{L}_{\mathrm{focal}}$ can be formally defined as:

$$\mathcal{L}_{\mathrm{focal}} = -\frac{1}{|\mathcal{V}_l|} \sum_{i \in \mathcal{V}_l} \sum_{c=0}^{C} \alpha_c \, y_{ic} \, (1 - \hat{y}_{ic})^{\gamma} \log(\hat{y}_{ic}), \tag{5}$$

where $\mathcal{V}_l$ is the node sets of labeled nodes, $\hat{y}_{ic}$ is the prediction score of node $v_i$ being classified as class $c$, $\gamma$ is the focusing parameter to control the rate at which easy nodes will be down-weighted, and $\alpha_c \in [0, 1]$ is a weighting hyper-parameter for different classes. Noted that $\mathcal{L}_{\mathrm{focal}} = \mathcal{L}_{\mathrm{ce}}$ when $\gamma = 0$ and $\alpha = 1$.

# 5 Experiments

In this section, we first introduce datasets we use to evaluate CM-GCL and baselines models. Then we provide the detailed analysis to show the effectiveness of CM-GCL and its strong applicability to real-world graph datasets. In addition, the baseline settings are provided in Section B of the Appendix.

## 5.1 Experimental Setup

### 5.1.1 Dataset

In this paper, we adopt four multi-modality graph datasets from existing works, i.e., AMiner [36], YelpChi [33], GitHub [29], and Instagram [30], which contain the raw content (e.g., text or image) and the graph structure information. Specifically, **AMiner** graph is a paper-citation academic network with raw text content (i.e, title). We select a sub-graph (having 18,089

Table 1: The number of labeled nodes, classes, and the imbalance ratio (IR) for each dataset.

| Dataset | AMiner | YelpChi | GitHub | Instagram |
|---|---|---|---|---|
| # of label | 18,089 | 67,395 | 20,895 | 8,651 |
| # of class | 5 | 2 | 2 | 2 |
| IR | $\approx 1.0$ | 0.15 | 0.50 | 0.59 |

papers and 22,864 authors) from AMiner dataset, and aim to predict the domain label (i.e., data mining, medical informatics, theory, visualization, and database) of each paper. In order to solve a harder classification task, we cast aside venue nodes from AMiner academic network and consider two relation types (i.e., author-write-paper and author-collaborate-author) for this academic graph. Different relation types are regarded as the same type. Each paper is characterized by the bag-of-words of keywords. We also count the number of published papers for each author as the attribute feature. In this dataset, the class distribution is relatively balanced (Tabel 1), so we use an imitative imbalanced setting by randomly selecting two classes (two out of five) as the minority classes. **YelpChi** provided by [33] is a benchmark graph dataset with the raw reviews for a set of restaurants and hotels in Chicago. We utilize YelpChi dataset to identify genuine reviews (8,919) and fake reviews (58,476), which is a binary classification task. Similar to AMiner, each review is characterized by the bag-of-words of keywords. Followed by [33], we also extract three types of relations among reviews. **GitHub** graph, including 6,965 malicious and 13,930 benign repositories, aims to detect malicious repositories on GitHub platform, which contains the raw content of repositories. Please refer to the detailed graph information in [29]. To mimic the imbalanced situation, we employ an imitative imbalanced setting with different imbalance ratios. **Instagram** graph contains the image and text information about users and posts and the corresponding rich relationships. Different from other three datasets, we mainly utilize the image information within this dataset to pre-train our proposed model and further detect illicit drug traffickers (including 3,242 drug traffickers and 5,409 normal users) on Instagram.

### 5.1.2 Baseline Method

To evaluate the performance of CM-GCL, we compare CM-GCL with nine baseline methods which are divided into four groups (Table 2): feature-based method (G1), graph learning models (G2), methods against imbalanced data (G3), and GCL models (G4). For G1 (Feature), we feed the attribute feature vector to a two-layer MLP [25]. For G2, we implement three graph learning methods, i.e., GCN [18], GAT [37], and GraphSAGE [9], to learn the node representations. For G3, we implement three popular approaches, i.e., over-sampling [17], re-weighting [4], and GraphSMOTE [51], to handle the imbalanced node representations generated by GCN. GraphSMOTE is the extension of SMOTE [1] on graph data, which leverages node similarities to generate synthesized nodes and edges. For G4, we reproduce two GCL methods, i.e., GCC [31] and GraphCL [45]. To compare them with CM-GCL fairly, we fine-tune the pre-trained models with Focal loss $\mathcal{L}_{\text{focal}}$ as well.

### 5.1.3 Experimental Settings

All experiments are conducted under the environment of the Ubuntu 16.04 OS, plus Intel i9-9900k CPU, two GeForce GTX 2080 Ti Graphics Cards, and 64 GB of RAM. We train all methods for each graph with a fixed epoch. Besides, all methods are trained ten times, and the average performance multiplied by 100 on testing data is reported. We use 70% samples for training, 10% for validation, and the remaining 20% for testing. Following existing works in evaluating imbalanced class classification [1, 51], we adopt two metrics, i.e., Macro F1-score (F1) and AUC-ROC score

Table 2: Performance comparison of all methods on imbalanced datasets with different ratios.

| Datasets | Ratio | Metric | Feature | GCN | GAT | Graph-SAGE | Over-sample | Re-weight | Graph-SMOTE | GCC | GraphCL | CM-GCL |
|---|---|---|---|---|---|---|---|---|---|---|---|---|
| **AMiner** (Node $\Leftrightarrow$ Text) | $D_{0.01}$ | F1 | 50.16 | 54.89 | 54.71 | 52.87 | 55.24 | **58.28** | 56.08 | 55.43 | 55.27 | 56.73 |
| | | AUC | 88.68 | 91.99 | 92.25 | 89.05 | 91.23 | 93.20 | 90.81 | 93.25 | 93.51 | **95.04** |
| | $D_{0.1}$ | F1 | 53.62 | 55.67 | 55.45 | 53.95 | 56.34 | 69.11 | 65.43 | 72.35 | 73.51 | **74.78** |
| | | AUC | 92.12 | 92.95 | 92.24 | 92.34 | 93.78 | 94.88 | 94.05 | 95.21 | 94.01 | **96.42** |
| | $D_{\text{init}}$ | F1 | 75.28 | 77.34 | 77.83 | 75.51 | 79.40 | 79.46 | 79.37 | 84.57 | 84.78 | **85.59** |
| | | AUC | 94.53 | 95.09 | 95.38 | 94.53 | 95.49 | 95.86 | 96.04 | 96.25 | 97.14 | **98.02** |
| **YelpChi** (Node $\Leftrightarrow$ Text) | $D_{0.01}$ | F1 | 46.86 | 49.58 | 49.80 | 47.75 | 49.78 | 50.98 | **51.41** | 49.98 | 50.23 | 50.45 |
| | | AUC | 57.11 | 59.08 | 60.51 | 58.84 | 89.69 | 82.25 | **90.94** | 87.35 | 88.52 | 90.79 |
| | $D_{0.1}$ | F1 | 52.60 | 56.27 | 57.14 | 53.62 | 65.01 | 67.08 | 70.08 | 72.47 | 73.38 | **75.10** |
| | | AUC | 65.29 | 83.74 | 83.84 | 78.83 | 89.91 | 85.63 | 91.81 | 91.04 | 91.23 | **93.02** |
| | $D_{\text{init}}$ | F1 | 53.46 | 64.57 | 65.21 | 46.46 | 72.61 | 72.77 | 77.04 | 77.45 | 78.56 | **80.75** |
| | | AUC | 67.02 | 85.04 | 85.45 | 82.70 | 89.42 | 87.85 | 92.20 | 92.28 | 93.45 | **94.48** |
| **Github** (Node $\Leftrightarrow$ Text) | $D_{0.01}$ | F1 | 23.88 | 32.33 | 32.51 | 29.41 | 35.73 | 37.77 | **45.97** | 42.51 | 43.20 | 45.89 |
| | | AUC | 57.91 | 80.09 | 79.25 | 77.87 | 82.97 | 84.66 | 86.73 | 85.54 | 86.21 | **88.51** |
| | $D_{0.1}$ | F1 | 64.92 | 69.15 | 69.42 | 66.14 | 71.64 | 72.45 | 74.43 | 72.51 | 73.24 | **76.35** |
| | | AUC | 79.69 | 86.85 | 86.25 | 84.25 | 89.76 | 91.48 | 93.37 | 91.54 | 92.39 | **95.27** |
| | $D_{\text{init}}$ | F1 | 66.74 | 72.55 | 73.27 | 70.48 | 73.18 | 74.12 | 77.59 | 79.54 | 80.59 | **83.35** |
| | | AUC | 78.21 | 88.28 | 88.24 | 86.82 | 90.99 | 92.84 | 93.32 | 92.74 | 93.75 | **96.48** |
| **Instagram** (Node $\Leftrightarrow$ Image) | $D_{0.01}$ | F1 | 4.35 | 12.51 | 12.58 | 10.29 | 15.28 | 17.39 | **23.48** | 19.32 | 20.42 | 22.47 |
| | | AUC | 51.04 | 58.71 | 58.91 | 56.27 | 60.24 | 62.75 | 66.41 | 64.05 | 64.28 | **67.34** |
| | $D_{0.1}$ | F1 | 36.02 | 42.62 | 42.32 | 40.54 | 44.11 | 47.53 | 49.65 | 48.32 | 49.88 | **51.24** |
| | | AUC | 62.71 | 81.45 | 81.98 | 79.27 | 83.83 | 85.24 | 88.42 | 85.32 | 87.21 | **89.14** |
| | $D_{\text{init}}$ | F1 | 66.77 | 72.55 | 72.32 | 71.24 | 73.05 | 74.57 | 75.32 | 77.25 | 78.25 | **80.71** |
| | | AUC | 76.83 | 87.28 | 87.12 | 85.39 | 88.68 | 89.05 | 89.24 | 91.08 | 92.53 | **94.25** |

(AUC) to evaluate all models. Besides, except for the initial imbalance ratio of datasets, to evaluate the model performance in more imbalanced scenarios, we set $\beta$ (imbalance ratio) as 0.1 and 0.01 and further utilize $D_{\text{init}}$, $D_{0.1}$, and $D_{0.01}$ to denote dataset with initial imbalance ratio, 0.1, and 0.01, respectively. Note that, we leverage $D_{\text{init}}$ to train CM-GCL and fine-tune the pre-trained encoder over $D_{\text{init}}$, $D_{0.1}$, and $D_{0.01}$ to conduct the downstream classification tasks. With the grid search, pruning ratio $e$ is set as 20%, the number of contrastive pairs $R$ for each node in intra-modality GCL is different for different graphs (e.g., 5 for AMiner graph), and the number of mini-batch $n$ is different for different tasks, (e.g., 100 for AMiner graph). Besides, the temperature parameter $\tau_{\text{inter}}$ and $\tau_{\text{intra}}$ are set as 0.1 and the trade-off hyper-parameter $\lambda$ among co-modality GCL is set as 0.5. For fine-tuning, $\alpha$ and $\gamma$ in $\mathcal{L}_{\text{focal}}$ for different graphs are different (e.g., (0.75, 1.0) for AMiner graph).

## 5.2 Performance Comparison

Table 2 shows the performances of all models on four graphs with different imbalance ratios. The best performances are highlighted in bold and the second-best performances are emphasized by underline. In Table 2, we can conclude that: (i) Merely considering content data (Feature) is not supportive enough to learn node representations. Integrating rich content and relations in graph can learn better representations. (ii) Leveraging unlabeled data to train graph encoder via contrastive learning can enhance node representations (i.e., GCC and GraphCL). Note that, to compare with baseline models fairly, most models (i.e., over-sampling, re-weighting, GraphSMOTE, GCC, GraphCL, and CM-GCL) are based on the embeddings generated from a two-layer GCN. (iii) Some popular methods (i.e., over-sampling, re-weighting, and GraphSMOTE) against imbalanced data in supervised learning can alleviate the influence of imbalance and further improve the performance. In addition, GraphSMOTE has excellent performance when the class is too imbalanced (i.e., $D_{0.01}$). (iv) CM-GCL has excellent

performance in most scenarios when dealing with imbalanced node classification over four real-world graphs. By comparing CM-GCL with GCL models (i.e., GCC and GraphCL), we also conclude that CM-GCL can learn better representations for real-world graphs.

## 5.3 Ablation Study

To show the effectiveness of different components in CM-GCL, we conduct a set of ablation experiments over subset $D_{0.1}$ of AMiner (node-text) and Instagram (node-image) datasets and further analyze the contribution of each component (i.e., co-modality GCL, pruning, and intra-modality GCL) by removing it separately. First, we remove co-modality GCL from CM-GCL, which means we directly employ a two-layer GCN to learn the node embedding for both datasets. As shown in Table 3, we can conclude that our model is effective as the performances on both datasets drop significantly. In addition, we remove network pruning and intra-modality

Table 3: Comparison among model variants of CM-GCL on $D_{0.1}$.

| Dataset | Method | F1 | AUC |
|---|---|---|---|
| **AMiner** (Node ⇔ Text) | CM-GCL | **74.78** | **96.42** |
| | - Co-Modality | 56.21 | 92.34 |
| | - Pruning | 72.42 | 95.31 |
| | - Intra-Modality | 72.51 | 95.23 |
| **Instagram** (Node ⇔ Image) | CM-GCL | **51.24** | **89.14** |
| | - Co-Modality | 42.62 | 81.45 |
| | - Pruning | 48.53 | 87.18 |
| | - Intra-Modality | 49.25 | 87.69 |

on CM-GCL. We find the performance on both datasets decreases obviously, validating the effectiveness of network pruning and intra-modality in CM-GCL.

## 5.4 CM-GCL Uncovers More Minority Samples

We further investigate whether network pruning is effective to discover minority samples during contrastive learning. Based on the pruning process introduced in Section 4.2.2, we measure the distribution of PIEs mined by CM-GCL over the different subsets (i.e., $D_{init}$, $D_{0.1}$, $D_{0.01}$) of Instagram (node-image) graph. Specifically, we sample the top 1% training data that are most easily affected by pruning, and further calculate the percentages of PIEs that belong to the minority class. As shown in Figure 2, minority samples in $D_{0.1}$ and $D_{0.01}$ are more easily to be detected by CM-GCL along with the training epochs. Notwithstanding the minority samples

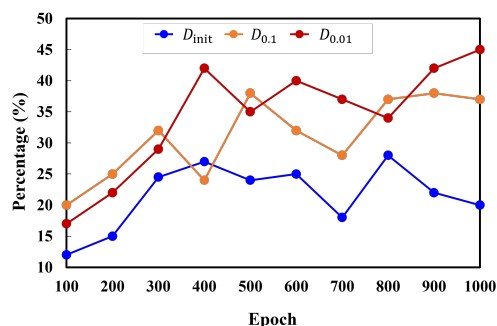

Figure 2: The percentage of PIEs that belong to minority classes under different training epochs.

in $D_{0.01}$ are less likely to be detected at the beginning, it shows a general increasing trend along with training epochs. $D_{init}$ keeps a relatively low percentage during the pre-training process. Hence, we again demonstrate that network pruning in CM-GCL is able to mine minority samples and further learn more balanced representations.

## 5.5 CM-GCL is Applicable to Mainstream Models in Different Modalities

Our proposed CM-GCL aims to study various real-world graphs: text-based graphs (i.e., AMiner, YelpChi, and GitHub), image-based graphs (i.e., Instagram). To this end, CM-GCL is designed as a plug-and-play tool that is applicable to most mainstream models in different modalities. Hence, we adopt at least two models in each modality (i.e., graph: GCN, GAT, and GraphSAGE (SAGE); text: BERT and DistilBERT (DBERT); image: ResNet50 (ResNet) and Swin Transformer (ST)) in Figure 3 to show the strong applicability and the effectiveness of

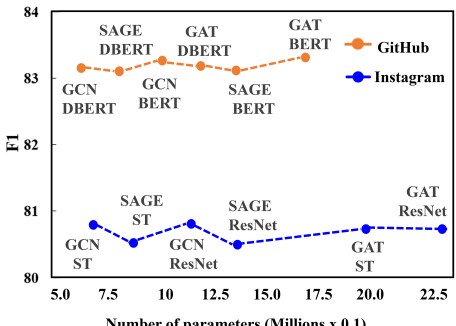

Figure 3: Model size vs. F1 score over $D_{init}$.

CM-GCL. This figure shows the model size and the performance (F1) of encoder combinations over the subset $D_{init}$ of GitHub (node-text) and Instagram (node-image) graphs. We find that all combinations have excellent performances. Due to efficiency and resource concerns, We choose GCN as the graph encoder, Swin Transformer as the image encoder, and DistilBERT as the text encoder for our experiments. In addition, besides the above models in Figure 3, our framework is applicable to other mainstream models.

### 5.6 Hyper-parameter Sensitivity

To explore the hyper-parameter sensitivity of CM-GCL, we conduct four sets of experiments w.r.t. the number R in Eq. 2 of positive node pairs for each node in intra-modality GCL, the trade-off hyper-parameter $\lambda$ in Eq. 4 between inter-modality GCL and intra-modality GCL, the percent $e$ of pruning parameters for co-modality GCL, and the hyper-parameters $(\alpha, \gamma)$ in $\mathcal{L}_{focal}$ (Eq. 5) for the downstream class-imbalanced node classification. Note that, the hyper-parameter sensitivity analysis about $R$ (Figure 4.(a)) and $\lambda$ (Figure 4.(b)) are based on AMiner and Instagram graphs with the initial imbalance ratio ($D_{init}$). The analysis in terms of the class imbalance problem is conducted on the dataset $D_{0.1}$ of AMiner and Instagram graphs (Figure 4.(c) and Figure 4.(d)) . Specifically, in Figure 4.(a), we vary the value of $R$ in the range

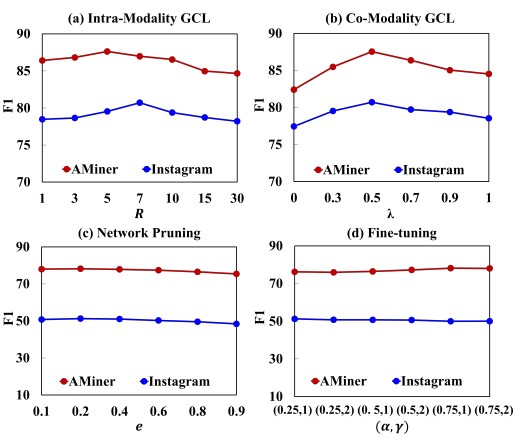

Figure 4: Hyper-parameters sensitivity analysis on two datasets.

of $\{1, 3, 5, 7, 10, 15, 30\}$ to generate positive samples for each node in intra-modality GCL. We can find that the optimal value for AMiner graph and Instagram graph is 5 and 7, respectively, while the performance drops obviously with too many positive pairs. That is to say, the intra-modality GCL has inverse performance if the positive pairs are not qualified enough. Besides, in Figure 4.(b), we vary the trade-off value $\lambda$ during pre-training. By comparing the performance when $\lambda = 0$ and $\lambda = 0.5$ (optimal value), we can demonstrate the effectiveness of inter-modality GCL in enhancing the performance of CM-GCL. Meanwhile, by comparing the performance when $\lambda = 0.5$ and $\lambda = 1.0$, we also validate the effectiveness of intra-modality GCL in CM-GCL. In addition, from Figure 4.(c), we vary the pruning percent $e$ in the range of $\{0.1, 0.2, 0.4, 0.6, 0.8, 0.9\}$ to prune encoders' network. We conclude that network pruning is effective for CM-GCL, while the performance will drop if the percent is large (i.e., 0.9). Lastly, we can conclude from Figure 4.(d) that the optimal value of $(\alpha, \gamma)$ in $\mathcal{L}_{focal}$ is (0.75,1.0) for AMiner graph and (0.25,1.0) for Instagram graph.

## 6  Conclusion

In this work, we develop a graph contrastive learning model (CM-GCL) to handle real-world multi-modality graphs with the class imbalance problem. Specifically, we propose co-modality GCL including inter-GCL and intra-GCL to automatically generate contrastive pairs based on the rich content. In addition, we propose network pruning to uncover minority samples during co-modality GCL pre-training. Our proposed CM-GCL is designed as a plug-and-play tool that is applicable to most mainstream models in different modalities. Extensive experiments across real-world graph datasets demonstrate that CM-GCL outperforms most state-of-the-art baselines for multiple downstream tasks and can alleviate the data imbalance problem.

## Acknowledgments

This work is partially supported by the NSF under grants IIS-2209814, IIS-2203262, IIS-2214376, IIS-2217239, OAC-2218762, CNS-2203261, CNS-2122631, CMMI-2146076, and the NIJ 2018-75-CX-0032. Any opinions, findings, and conclusions or recommendations expressed in this material are those of the authors and do not necessarily reflect the views of any funding agencies.

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
