# Co-Modality Graph Contrastive Learning for Imbalanced Node Classification -Appendix

**Yiyue Qian[1], Chunhui Zhang[2], Yiming Zhang[3], Qianlong Wen[1],**
**Yanfang Ye[1*], Chuxu Zhang[2*]**

yqian5@nd.edu, chunhuizhang@brandeis.edu, yxz2092@case.edu,
qwen@nd.edu, yye7@nd.edu, chuxuzhang@brandeis.edu

[1]Department of Compute Science and Engineering, University of Notre Dame, USA
[2]Department of Computer Science, Brandeis University, USA
[3]Department of Computer and Data Sciences, Case Western Reserve University, USA

In this appendix, we first provide the pseudo-code of CM-GCL to facilitate the understanding of CM-GCL. We then discuss the experimental settings of baseline methods.

## A  Pseudo-code of CM-GCL

In CM-GCL, we can either take the text feature $x_T$ or the image feature $x_I$ as the content feature, and consider the corresponding text encoder $f_T$ or image encoder $f_I$ as the content encoder. In Algorithm 1, we take text feature $x_T$ and the corresponding text encoder $f_T$ as an example.

---
**Algorithm 1:** Training Procedure of CM-GCL

**Data:** Graph $G$, Node feature $x_G$, Content feature $x_T$, GNN encoder $f_G$, Content encoder $f_T$,
     Node similarity set $\mathcal{S}$

**Result:** $f_G^*$: pre-trained GNN encoder; $f_T^*$: fine-tuned content encoder

1   Initialize $(f_G, f_T)$ with parameters $(\theta_G^0, \theta_T^0)$.

2   **for** *each epoch* **do**

3      **for** *each step $l$* **do**

4          Prune $e\%$ of smallest-magnitude parameters in $(\theta_G^l, \theta_T^l)$ by creating the corresponding
            masks $(m_G^l, m_T^l)$.

5      Obtain the embedding $(\widetilde{h}_G, \widetilde{h}_T)$ from $(\widetilde{f}_G, \widetilde{f}_T)$; Project: $(\widetilde{h}_G, \widetilde{h}_T) \rightarrow (\widetilde{z}_G, \widetilde{z}_T)$.

6      Calculate the inter-modality GCL loss $\mathcal{L}_{\text{inter}}$ in Eq. 1 .

7      Release masks $(m_G, m_T)$.

8      **for** *each step $l$* **do**

9          Prune $e\%$ of smallest-magnitude parameters in $\theta_G^l$ by creating corresponding masks $m_G^l$.

10     Obtain the embedding $(h_G, \widetilde{h}_G)$ from $(f_G, \widetilde{f}_G)$; Project: $(h_G, \widetilde{h}_G) \rightarrow (z_G, \widetilde{z}_G)$.

11     Calculate the intra-modality contrastive loss $\mathcal{L}_{\text{intra}}$ in Eq. 3 among $(z_G, \widetilde{z}_G)$.

12     Release masks $m_G$.

13     Optimize $f_G, \widetilde{f}_G, \widetilde{f}_T$ by minimizing the co-modality graph contrastive loss $\mathcal{L}$ in Eq. 4.

14 **return** $f_G^*$, $f_T^*$.

---

---
*Corresponding authors.

36th Conference on Neural Information Processing Systems (NeurIPS 2022).

# B   Baseline Settings

In this section, we discuss the settings of baseline models for imbalanced node classification over four graphs.

**G1:** We convert the rich text content into the bag-of-words feature vectors, and further feed the feature vectors with different imbalance ratios to a two-layer MLP [7] classifier to get the classification results. For AMiner, YelpChi, and GitHub graph datasets, we implement CHI-Square [11] to select useful feature words. For Instagram graph, we employ ResNet50 [3] to obtain the initial image representations as the feature vector.

**G2:** We implement three graph neural network based representation learning models including GCN [5], GAT [9], and GraphSAGE [2] to learn the node embeddings by leveraging both node feature (bag-of-words feature vector) and graph structure information. Specifically, the number of layers in GCN, GAT, and GraphSAGE is set as 2. In addition, the number of attention heads in GAT is set as 8. Node representations generated by graph representation learning models are fed into a two-layer MLP classifier. Except for the dataset $D_{\text{init}}$ with initial imbalance ratio, we also fed the imbalanced data (i.e., $D_{0.1}$ and $D_{0.01}$) to train the GNN models and further get the classification over imbalanced testing nodes.

**G3:** We also implement three popular approaches against imbalanced datasets, i.e., over-sampling [4], re-weighting [1], and GraphSMOTE [12]. For over-sampling and re-weighting methods, they handle the node representations generated by a two-layer GCN and the node representations are further fed to a two-layer MLP. Specifically, we over-sample the nodes belonging to the minority classes by adding the duplicated nodes and the duplicated edges among nodes (over-sampling) or we emphasize the weight for minority classes during optimization (re-weighting). For GraphSMOTE, we utilize the similarities among nodes to synthesize the nodes in monitory classes and train the edge generator to learn relationships among nodes simultaneously. Different from the setting in GraphSMOTE, we employ a two-layer GCN as the feature extractor such that we compare GraphSMOTE with other baseline models fairly.

**G4**: To compare the ability of representation learning of CM-GCL, we conduct two graph contrastive learning models, i.e., GCC [8] and GraphCL [10], to pre-train the graph encoder. To compare CM-GCL with these aforementioned graph contrastive learning models fairly, all encoders are set as a two-layer GCN. For GCC, we adopt an end-to-end strategy to build the subgraph dictionary (with size 1023) in contrastive learning. For GraphCL, we adopt edge perturbation and node dropping as graph augmentation methods to generate contrastive pairs and further pre-train the graph encoder during contrastive learning. For the downstream classification, similar to CM-GCL, we fine-tune the GCN encoder pre-trained by GCC or GraphCL and further adopt Focal loss [6] to handle node classification over imbalanced datasets.