# OpenReview forum: "Co-Modality Graph Contrastive Learning  for Imbalanced Node Classification"
_NeurIPS.cc/2022/Conference — NeurIPS 2022 Accept_

### Official Review · Reviewer_D25V · 2022-07-07

**Rating:** 5
**Confidence:** 4
**Soundness:** 1 poor
**Presentation:** 1 poor
**Contribution:** 1 poor

**Summary:**

This paper introduces a model to learn representations over graph nodes such that: (1) the representations can address imbalanced nature of classes, and (2) they use multimodality to learn representations on nodes. The learned representations show good performance compared to baselines in various settings on two benchmarks. As the authors mention, the limit of their approach is that it assumes corresponding textual or visual data is available.

**Questions:**

- How would your model perform in case that some corresponding modality is missing from some nodes?
- How would the model perform on other publicly available benchmarks?
- How you would use your model for single modality graph data?

**Ethics Review Area:**

["I don’t know"]

**Strengths And Weaknesses:**

Strengths:
- The paper tackles the problem of imbalanced data in node-level representation learning while assuming that corresponding modalities are available. This setting can be important in some real-world graph data such as social networks.
- The ideas in the paper are easy-to-follow.
Weaknesses
- The paper needs some revisions to address typos, e.g., "Existing graph representation learning methods [9, 41, 29] generally implement the pre-trained language model (e.g., BERT [6])".
- Experimental results are carried over only two datasets (Github and Instagram) which is not enough. There are other multimodal graph benchmarks publicly available that authors could have used.
- Important citations are missing from GCL literature are missing [1, 2, 3, 4]

[1] https://arxiv.org/abs/2006.05582
[2] https://arxiv.org/abs/2201.09830
[3] https://arxiv.org/abs/2106.07594
[4] https://arxiv.org/abs/2102.06514

---

> ### Author Response · Authors · 2022-08-02
> **Response to Reviewer D25V (2/2)**
>
> Thank you for your comments and we will address your concerns about applying CMI-GCL to more benchmark datasets in detail.
>
> Q6: Apply CMI-GCL to more benchmark datasets.
>
> **A6**: Thank you for your suggestions. CMI-GCL focuses on utilizing the rich content information to co-train encoders in multiple modalities for real-world imbalanced graph data. Due to privacy or other issues, most graph benchmark datasets merely provide the processed feature vector (e.g.,  ogbn-product, ogbn-arxiv, Cora, and BlogCatalog). To make our model more convincing, we apply CMI-GCL to another benchmark data with raw text content: AMiner paper-citation graph [1]. We select a dataset from AMiner, having  18,090 papers and 29,411 authors, to predict the domain label of each paper. As the class distributions are relatively balanced, similar to the multi-class imbalanced setting in GraphSMOTE [2], we use an imitative imbalanced setting that two random classes are selected as minority classes and further down-sampled nodes in minority classes. Similar to the setting of data splitting (70\% for model training, 10\% for model validation, and 20\% for model testing) in GitHub and Instagram datasets, each majority class has a training set of 2,500 nodes, while the number of training nodes for each minority class is 2,500 $\times$ $\beta$ (0.01 and 0.1). Here $\beta$ is the imbalance ratio under different imbalanced scenarios.
>
> The following Table 1 shows the performance of CMI-GCL and four excellent baseline models (based on the performance on GitHub and Instagram datasets ). Due to the time limit of rebuttal, partial results are provided but we will finish experiments on other baseline methods in the final paper.  We can see that CMI-GCL has the best performance under different imbalanced scenarios by comparing it with GCN, GraphSMOTE, GCC+Focal, and GraphCL+Focal. Specifically, CMI-GCL outperforms the other two  GCL models (i.e., GCC and GraphCL), showing that CMI-GCL can learn better representations in imbalanced datasets. Besides, CMI-GCL has better performance than GraphSMOTE [2], the specific SMOTE model for dealing with imbalanced graph datasets.
>
> $\textbf{Table 1}:$ Performance comparison on AMiner dataset under different imbalanced ratios.
>
> | Model| $D_{0.01}$-F1|$D_{0.01}$-AUC|$D_{0.1}$-F1|$D_{0.1}$-AUC|$D_{init}$-F1|$D_{init}$-AUC|
> | ----------- | ----------- |----------- |----------- |----------- |----------- |----------- |
> | GCN|21.71 ±2.71|  76.48 ±2.59    | 74.41  ±2.31    | 85.59  ±2.38    | 77.45 ±2.27     |  87.23 ±2.19    |
> | GraphSMOTE| 28.42 ±1.41      | 81.74  ±1.34    | 77.95 ±1.57      |  90.14 ±1.75    | 90.53 ±1.84    |  91.55 ±1.92   |
> | GCC+Focal | 26.41  ±1.64      | 80.37 ±1.53    | 77.56  ±1.49      |  90.39 ±1.24    | 89.94 ±1.38      |  92.05 ±1.04    |
> | GraphCL+Focal|27.36 ±1.55      |  81.25  ±1.47    | 78.28  ±1.07      |  91.47±1.12    |91.28 ±0.94     | 93.07 ±0.81   |
> |**CMI-GCL**| $\textbf{32.52}$ ±0.97 | $\textbf{86.37}$ ±0.93  | $\textbf{83.97}$ ±0.83 |  $\textbf{96.82}$ ±0.75 | $\textbf{96.85}$ ±0.74 | $\textbf{98.45}$ ±0.65 |
>
> Moreover, we also fine-tune the pre-trained GNN model from CMI-GCL on AMiner data in freezing mode to evaluate the balancedness of representations. The following Table 2 shows the balancedness comparison of CMI-GCL, GCC, and GraphCL on $D_{b}$, $D_{init}$, $D_{0.1}$, and $D_{0.01}$. Balancedness results of SimCLR and HeCo will be provided in the final paper. From Table 2, we can conclude that CMI-GCL can learn more balanced representations over heavily imbalanced data ($D_{0.01}$) by comparison with GCC and GraphCL. In addition, CMI-GCL still has excellent balancedness performance in relatively imbalanced data $D_{0.1}$, while the performance of GCC and GraphCL drops obviously in $D_{0.1}$. Please refer to Section E in the revised Appendix file for more details.
>
> $\textbf{Table 2}:$ Balancedness comparison of contrastive models learned on $D_{b}$, $D_\text{init}$, $D_\text{0.1}$, and $D_\text{0.01}$ of Aminer Data.
>  |Ratio|CMI-GCL|GCC|GraphCL|
> | ----------- |----------- |----------- |----------- |
> |$D_{b}$|$\textbf{49.71}$ ±0.53 | 49.52 ±1.05| 49.70 ±1.21|
> |  $D_\text{init}$ | $\textbf{49.55}$ ±0.57 | 47.31 ± 1.24 | 48.41 ± 1.27|
> |$D_{0.1}$     |$\textbf{48.57}$ ±0.64|   45.38 ± 1.37| 45.71 ± 1.39   |
> |$D_{0.01}$     |  $\textbf{47.63}$ ± 0.69  | 39.51 ± 1.45 | 39.57 ± 1.37   |
>
>
> [1] AMiner Dataset, https://www.aminer.org/data/, 2012.
>
> [2] Graphsmote: Imbalanced node classification on graphs with graph neural networks, WSDM'21.

---

> > ### Comment · Reviewer_D25V · 2022-08-08
> > **Thank you**
> >
> > Thanks for the response. Following your answers and also answers to other reviewers I increased the score.

---

> > > ### Author Response · Authors · 2022-08-09
> > > **Additional response to Reviewer D25V**
> > >
> > > Thank you very much for increasing your rating. We try to address your concerns during the rebuttal and discussion period. To further make CMI-GCL more convincing, except for the AMiner dataset, we apply CMI-GCL to another benchmark dataset with raw text: YelpCHI [1], to detect spam reviews on Yelp (binary classification).
> > >
> > > The number of reviews provided by [1] is 67,395 including 8,916 spam reviews and 58,479 normal reviews. We need to mention that we not only utilize the raw review text as the input to co-train encoders in both graph-modal (e.g., GCN) and text-modal (e.g., DistilBERT), but also consider the feature vector (32 dimensions) provided by [1] in GNN models as these feature vectors describe the properties of nodes. Similar to the imbalance setting of GitHub, Instagram, and AMiner, we have three imbalanced subsets $D_{0.01}$, $D_{0.1}$, and $D_{\text{init}}$ (init = 0.15), and the following table shows the performance of CMI-GCL and three competitive baseline models for spam review detection on YelpCHI dataset. We will finish other baseline experiments in the final paper.  From Table 1, we can conclude that CMI-GCL has excellent performance, especially in extremely imbalanced subsets ($D_{0.01}$), which again demonstrates the effectiveness of CMI-GCL in handling imbalanced graph datasets.
> > >
> > > $\textbf{Table 1}:$ Performance comparison on YelpCHI dataset under different imbalanced ratios.
> > >
> > > | Model| $D_{0.01}$-F1|$D_{0.01}$-AUC|$D_{0.1}$-F1|$D_{0.1}$-AUC|$D_{init}$-F1|$D_{init}$-AUC|
> > > | ----------- | ----------- |----------- |----------- |----------- |----------- |----------- |
> > > | GCN| 65.31 ±3.63| 69.37 ±3.52  |73.25  ±3.24 | 76.25  ±3.41| 75.45 ±3.21| 78.52 ±3.42
> > > | GraphSMOTE| 74.51 ±1.34 |78.15  ±1.31|79.97 ±1.38 |82.59 ±1.32| 81.42 ±1.31| 84.57 ±1.27   |
> > > | GraphCL+Focal|73.24 ±1.37 | 77.31  ±1.35|78.41 ±1.37 | 81.07 ±1.24| 80.57 ±1.21 | 83.39 ±1.18   |
> > > |**CMI-GCL**| $\textbf{78.58}$± 0.61 | $\textbf{82.25}$ ±0.67  | $\textbf{83.45}$ ±0.54| $\textbf{86.46}$ ± 0.62| $\textbf{84.25}$ ± 0.53| $\textbf{87.54}$ ± 0.48 |
> > >
> > > Thanks again for your efforts in reviewing our work and we hope our response can address your concerns. If so, we really appreciate it if you can accept our work.
> > >
> > >  [1] Collective Opinion Spam Detection: Bridging Review Networks and Metadata, KDD'15.

---

> ### Author Response · Authors · 2022-08-02
> **Response to Reviewer D25V (1/2)**
>
> Q1: How will CMI-GCL perform when the corresponding modality of some nodes is missing?
>
> **A1**: We use three datasets in this work, i.e, GitHub, Instagram, and AMiner. GitHub and AMiner datasets do not have this issue as repositories in GitHub and papers in AMiner must have text content, while a few nodes in Instagram dataset miss the image content. In this scenario, we select nodes with images to pre-train CMI-GCL and further fine-tune the pre-trained GNN on all nodes. The small proportion of nodes (21 out of 88,356) missing image content are given random vectors for model fine-tuning.
>
> Q2: How will CMI-GCL perform on other public benchmark datasets?
>
> **A2**: Our proposed model CMI-GCL is applicable to various real-world imbalanced graph datasets. Unlike existing GCL models that need too many resources to find optimal graph augmentation methods for contrastive pairs under various real-world scenarios, CMI-GCL generates contrastive pairs automatically based on the rich content information. Besides, to better handle various real-world graph datasets, our model CMI-GCL is designed as a plug-and-play tool that integrates most mainstream models in different modalities.
>
> Due to privacy or other issues, most benchmark datasets release the constructed graphs with structure data and processed feature vectors. For these benchmark datasets with raw text, CMI-GCL can handle these benchmark datasets with excellent performance (please refer to **A6** for the experiments on a benchmark dataset AMiner). For most benchmark graph datasets without raw content, as we will discuss in **A3**,  CMI-GCL still works under different imbalance settings. Specifically, we can utilize the intra-modality GCL module in CMI-GCL to conduct contrastive learning by co-training the prune (sparse) GNN encoder and the non-pruned (dense) GNN encoder. Please refer to Section 4.2.3 for more details.
>
> Q3: How will CMI-GCL deal with single modality graph data? (Question 3)
>
> **A3**: For graph datasets with a single modality (e.g., graph structure), CMI-GCL still works with excellent performance according to the ablation study in Table 3 of the paper. Specifically, for a single modality graph data, we can conduct the intra-modality GCL module to pre-train the GNN model on the benchmark dataset. As discussed in Section 4.2.3, we can co-train the sparse GNN and dense GNN to conduct contrastive learning. Also in Table 3 of this paper, the ablation study shows that intra-modality GCL contributes to CMI-GCL on multiple datasets.
>
>
> Q4: Some typos.
>
> **A4**: Thank you for the suggestions. We have fixed these typos in the revised paper.
>
> Q5: Cite some papers in GCL literature.
>
> **A5**: Thank you for your suggestion and we have cited these works in the related works section.

---

> ### Comment · Area_Chair_Cvh9 · 2022-08-08
> **To reviewers: please feedback to the authors' response if needed**
>
> We are approaching the end of the rebuttal period, and the authors have provided feedback to which your further response is appreciated, especially seeing there are dispute on the rating among the reviewers.

---

### Official Review · Reviewer_UmCM · 2022-07-11

**Rating:** 5
**Confidence:** 4
**Soundness:** 3 good
**Presentation:** 3 good
**Contribution:** 3 good

**Summary:**

The paper explored tackling data imbalance problem on graph with contrastive learning. This work proposes a principled framework called Co-Modality Imbalanced Graph Contratsive Learning with Network Pruning (CMI-GCL) to generate contrastive pairs and further learn balanced representation over unlabeled data. Inter- and intra-modality GCL are designed to learn general node representation. Network pruning is used to detect minority nodes. Experiments are conducted on two experiments to verify the effectiveness.

**Questions:**

Please refer to the weak points I mentioned above.

**Ethics Review Area:**

["I don’t know"]

**Limitations:**

The limitations are discussed in the conclusion. I believe the authors can find more related datasets at OGB benchmark. I’m not aware of any potential negative societal impact.

**Strengths And Weaknesses:**

For the strengths, the paper tackles the important data imbalance problem on graph data and designs effective co-modality contrastive learning strategy to learn multimodal representation. Extensive ablation study demonstrates the effectiveness of the proposed method. However, I'm confused about why network pruning can resolve the problem of data imbalance. The paper mentioned minority samples can be “forgotten” by pruning deep neural networks. But why the paper said pruning and releasing weights in each round can effectively boosts minority samples’ weights in the section "Network Pruning for Imbalance Data"? For the experiments, despite the datasets you used, OGB node property benchmark provides numerous large-scale graph datasets with raw text. Current experiments are only conducted on two datasets, which seems not very convincing. Moreover, it'll be better if you could provide the standard deviation of the experiments.

There are also some minor issues. In the abstract, the paper said "Hence, we co-train two pruned encoders ...". What is the causal relationship here? In both Figure 1 (above "Content Encoder") and 3, there are some weird gray lines, please consider removing them in the future revision.

---

> ### Author Response · Authors · 2022-08-02
> **Response to Reviewer UmCM (2/2)**
>
> Thank you for your valuable comments and we will answer your questions about applying CMI-GCL to more datasets in detail.
>
> Q5: Conduct experiments on more datasets.
>
> **A5**: Thank you for your suggestions. Most benchmark datasets on OGB platform [1] (e.g., ogbn-product, ogbn-arxiv, ogbn-papers100M, and ogbn-mag) provide the processed feature vectors rather than the raw content (e.g., text) due to privacy or other issues, while our model CMI-GCL focuses on utilizing the rich content information to co-train encoders in multiple modalities for real-world imbalanced graph data. To make our model more convincing, we apply CMI-GCL to another benchmark data with raw text content: AMiner paper-citation graph [2]. We select a dataset from AMiner which has 18,090 papers and 29,411 authors. The task is to predict the domain label of each paper. As the class distributions are relatively balanced, similar to the multi-class imbalanced setting in GraphSMOTE [3], we use an imitative imbalanced setting that two random classes are selected as minority classes and further down-sampled nodes in minority classes. Similar to the setting of data splitting (70\% for model training, 10\% for model validation, and 20\% for model testing) in GitHub and Instagram datasets, each majority class has a training set of 2,500 nodes, while the number of training nodes for each minority class is 2,500 $\times$ $\beta$ (0.01 and 0.1). Here $\beta$ is the imbalance ratio under different imbalanced scenarios. The following Table 1 shows the performance of CMI-GCL and four excellent baseline models (based on the performance on GitHub and Instagram datasets ). Due to the time limit of rebuttal, partial results are provided but we will finish experiments on other baseline methods in the final paper.
>
> From Table 1, we can conclude that CMI-GCL has the best performance under different imbalanced scenarios. Specifically, by comparing CMI-GCL with GCN, GCC+Focal, and  GraphCL+Focal, we find that our designed pre-training model CMI-GCL, including inter-modality pruning GCL and intra-modality pruning GCL, can learn better representations for downstream tasks on AMiner data. Also, we can conclude that CMI-GCL outperforms GraphSMOTE [3], the specific SMOTE model against imbalanced graph datasets.
>
>
> $\textbf{Table 1}:$ Performance comparison on AMiner dataset under different imbalanced ratios.
>
> | Model| $D_{0.01}$-F1|$D_{0.01}$-AUC|$D_{0.1}$-F1|$D_{0.1}$-AUC|$D_{init}$-F1|$D_{init}$-AUC|
> | ----------- | ----------- |----------- |----------- |----------- |----------- |----------- |
> | GCN|21.71 ±2.71|  76.48 ±2.59    | 74.41  ±2.31    | 85.59  ±2.38    | 77.45 ±2.27     |  87.23 ±2.19    |
> | GraphSMOTE| 28.42 ±1.41      | 81.74  ±1.34    | 77.95 ±1.57      |  90.14 ±1.75    | 90.53 ±1.84    |  91.55 ±1.92   |
> | GCC+Focal | 26.41  ±1.64      | 80.37 ±1.53    | 77.56  ±1.49      |  90.39 ±1.24    | 89.94 ±1.38      |  92.05 ±1.04    |
> | GraphCL+Focal|27.36 ±1.55      |  81.25  ±1.47    | 78.28  ±1.07      |  91.47±1.12    |91.28 ±0.94     | 93.07 ±0.81   |
> |$\textbf{CMI-GCL}$| $\textbf{32.52}$ ±0.97 | $\textbf{86.37}$ ±0.93  | $\textbf{83.97}$ ±0.83 |  $\textbf{96.82}$ ±0.75 | $\textbf{96.85}$ ±0.74 | $\textbf{98.45}$ ±0.65 |
>
> Besides, similar to GitHub and Instagram datasets, we also fine-tune the pre-trained GNN model on AMiner data in freezing mode to evaluate the balancedness of learned representations. Table 2 shows the balancedness comparison of CMI-GCL, GCC, and GraphCL on $D_{b}$, $D_{init}$, $D_{0.1}$, and $D_{0.01}$. Balancedness results of SimCLR and HeCo will be completed in the final paper. From Table 2, we can conclude that CMI-GCL can learn more balanced representations over heavily imbalanced data ($D_{0.01}$) by comparison with GCC and GraphCL. Please refer to Section E in the revised Appendix file for more details.
>
> $\textbf{Table 2}:$ Balancedness comparison of contrastive models learned on $D_{b}$, $D_\text{init}$, $D_\text{0.1}$, and $D_\text{0.01}$ of Aminer Data.
>  |Ratio|CMI-GCL|GCC|GraphCL|
> | ----------- |----------- |----------- |----------- |
> |$D_{b}$|$\textbf{49.71}$ ±0.53 | 49.52 ±1.05| 49.70 ±1.21|
> |  $D_\text{init}$ | $\textbf{49.55}$ ±0.57 | 47.31 ± 1.24 | 48.41 ± 1.27|
> |$D_{0.1}$     |$\textbf{48.57}$ ±0.64|   45.38 ± 1.37| 45.71 ± 1.39   |
> |$D_{0.01}$     |  $\textbf{47.63}$ ± 0.69  | 39.51 ± 1.45 | 39.57 ± 1.37   |
>
> [1] Open graph benchmark: Datasets for machine learning on graphs, NeurIPS'20.
>
> [2] AMiner Dataset, https://www.aminer.org/data/, 2012.
>
> [3] Graphsmote: Imbalanced node classification on graphs with graph neural networks, WSDM'21.

---

> ### Author Response · Authors · 2022-08-02
> **Response to Reviewer UmCM (1/2)**
>
> Thank you for your comments, assessment, and valuable time. We will answer your questions and concerns as follows. If our response can address all your concerns, we will appreciate it if you can increase your score.
>
> Q1: How does network pruning resolve the problem of data imbalance?
>
> **A1**: As we discussed in Section 4.2.2, network pruning will "forget" minority samples. Specifically, the weight parameter of minority samples is small during the model training and it will be "forgotten" if we prune the network based on the magnitude pruning method. Magnitude pruning is one of the typical pruning techniques that utilize the absolute value of weights to rank their importance and remove weights that are below the defined threshold (20\% in this paper). Some methods remove these weights permanently, while we mask the weights of minority nodes temporarily via step 3 (in Section 4.2.2) and further recover the weights and receive the gradient updates via step 4. With step 3, due to masks on minority nodes, the corresponding contrastive loss during the feed-forward process increases. Afterward, according to step 4, we release the masked weight and also update the weights for minority nodes during the backpropagation. By repeatedly training, pruning, and releasing weights, our model can boost the weights of minority nodes dynamically and re-balance the contrastive loss on minority samples implicitly.
>
> Q2: Provide the standard deviation of experimental results.
>
> **A2**: Thank you for your suggestions. Due to the space limit, we just provided the mean value of ten runs in our experiments. To compare the performance among different methods more precisely, we add the standard deviation for GitHub data in Table 3 and Instagram data in Table 4 in the revised Appendix file (emphasized in red). Please refer to them for more details.
>
> Q3: What's the causal relationship in the sentence "Hence, we co-train two pruned encoders" in the abstract?
>
> **A3**: As mentioned in the previous sentence, we extend network pruning to our GCL framework to detect minority nodes. Then we explain how we integrate network pruning into our GCL framework. Following your suggestion, we change "hence" to "based on this" in the revised paper.
>
>
> Q4: Minor issues.
>
> **A4**: Thanks for the suggestions. We have fixed these minor issues in the revised paper.

---

> > ### Comment · Reviewer_UmCM · 2022-08-08
> > **Response to Authors**
> >
> > Thanks for the response. I'm still confused about the algorithmic process. Why masking partial weights of the **network** will lead to "masks on minority **nodes**", and then "contrastive loss during the feed-forward process increases"? Besides this, most of the statements in the text are assumptions, I think they need to be theoretically or empirically verified for this new setting. At present, only Figure 2 shows that the proposed method can detect some minority samples, which is not convincing enough for me. For the experiments, I appreciate the new experiments on a new data set. But for OGB benchmark, most data sets not only provide the processed feature vectors but also the raw content. You can check their github page or this github issue https://github.com/snap-stanford/ogb/issues/240. Due to above reasons, I stick to the borderline rating.

---

> > > ### Author Response · Authors · 2022-08-09
> > > **Additional response to Reviewer UmCM (2/2)**
> > >
> > > ***"For the experiments, I appreciate the new experiments on a new data set. But for the OGB benchmark, most data sets not only provide the processed feature vectors but also the raw content. You can check their GitHub page or this GitHub issue."***
> > >
> > > Thank you for pointing out the new data. We could not find the raw text from the official OGB website and they mention that they only provide the feature vector with 128 dimensions for each node [1]. As the discussion period almost ends, we do not have enough time for the experiments on the OGB paper-100M dataset, but we will apply CMI-GCL to this benchmark dataset in our final paper. To make CMI-GCL more convincing, we find another relatively small benchmark dataset with raw text: YelpCHI [2], and further apply CMI-GCL to detect spam reviews on Yelp (binary classification).
> > >
> > > The number of reviews provided by [2] is 67,395 including 8,916 spam reviews and 58,479 normal reviews. We need to mention that we not only utilize the raw review text as the input to co-train encoders in both graph-modal (e.g., GCN) and text-modal (e.g., DistilBERT), but also consider the feature vector (32 dimensions) provided by [2] in GNN models as these feature vectors describe the properties of nodes. Similar to the imbalance setting of GitHub, Instagram, and AMiner, we have three imbalanced subsets $D_{0.01}$, $D_{0.1}$, and $D_{\text{init}}$ (init = 0.15), and the following table shows the performance of CMI-GCL and three competitive baseline models for spam review detection on YelpCHI dataset. We will finish other baseline experiments in the final paper.  From Table 1, we can conclude that CMI-GCL has excellent performance, especially in extremely imbalanced subsets ($D_{0.01}$), which again demonstrates the effectiveness of CMI-GCL in handling imbalanced graph datasets.
> > >
> > > $\textbf{Table 1}:$ Performance comparison on YelpCHI dataset under different imbalanced ratios.
> > >
> > > | Model| $D_{0.01}$-F1|$D_{0.01}$-AUC|$D_{0.1}$-F1|$D_{0.1}$-AUC|$D_{init}$-F1|$D_{init}$-AUC|
> > > | ----------- | ----------- |----------- |----------- |----------- |----------- |----------- |
> > > | GCN| 65.31 ±3.63| 69.37 ±3.52  |73.25  ±3.24 | 76.25  ±3.41| 75.45 ±3.21| 78.52 ±3.42
> > > | GraphSMOTE| 74.51 ±1.34 |78.15  ±1.31|79.97 ±1.38 |82.59 ±1.32| 81.42 ±1.31| 84.57 ±1.27   |
> > > | GraphCL+Focal|73.24 ±1.37 | 77.31  ±1.35|78.41 ±1.37 | 81.07 ±1.24| 80.57 ±1.21 | 83.39 ±1.18   |
> > > |**CMI-GCL**| $\textbf{78.58}$± 0.61 | $\textbf{82.25}$ ±0.67  | $\textbf{83.45}$ ±0.54| $\textbf{86.46}$ ± 0.62| $\textbf{84.25}$ ± 0.53| $\textbf{87.54}$ ± 0.48 |
> > >
> > > Hope our response can address your concerns. If so, we really appreciate it if you can kindly increase your score. Please let us know if you have any more comments.
> > >
> > > [1] Open graph benchmark: Datasets for machine learning on graphs, NeurIPS'20.
> > >
> > >  [2] Collective Opinion Spam Detection: Bridging Review Networks and Metadata, KDD'15.

---

> > > ### Author Response · Authors · 2022-08-09
> > > **Additional response to Reviewer UmCM (1/2)**
> > >
> > >
> > > Thanks for your additional comments. We will reply to your additional questions/concerns as follows.
> > >
> > > ***"Most of the statements in the text are assumptions, I think they need to be theoretically or empirically verified for this new setting."***
> > >
> > > Thank you for the suggestion. The statement that PIE (Pruning Identified Exemplars) can be detected via network pruning has already been theoretically and empirically verified in previous works [1,2]. Specifically, [1]  systematically demonstrates that PIEs often show up at the long-tail of a distribution (minority samples). [2] also comprehensively demonstrates that minority images can be detected via pruning the network dramatically. In this paper, we employ the magnitude pruning method to prune the network for detecting minority nodes in the graph $G$.
> > >
> > > ***"Why masking partial weights of the network will lead to "masks on minority nodes", and then "contrastive loss during the feed-forward process increases"?"***
> > >
> > > As we mentioned in the previous response, the weight parameter of minority samples is always small during the model training [1]. In this case, we employ the magnitude pruning technique [3] to mask the weights of these minority samples that are below the user-specified threshold (20\%). For instance, give a node $v_i$ in $G$, if the absolute value of the weight parameter of $v_i$ is below the threshold during the model training, we will mask the weight parameter during the feed-forward process (step 3 in Section 4.2.2). In this case, due to masks on weight parameters, the representations of minority nodes generated by multi-modal encoders will be degraded, and the corresponding contrastive loss on minority nodes will further increase as the output of $v_i$ has large disagreements on the pseudo label we create for the inter-modality GCL module and intra-modality  GCL module.
> > >
> > > Afterwards, we release the masks on the weight parameters and further update the gradient of parameters for these minority samples during the backpropagation process (step 4). Thus, the parameters of these minority nodes will be boosted to update dynamically compared to the non-pruning network and the contrastive loss will be re-balanced on minority samples implicitly.
> > > The reason why some papers claim network pruning will `forget' minority samples is that they do not recover or release the masks during the backpropagation, while we employ the recovering mechanism [1] to detect minority samples and further boost the parameters updates during backpropagation.
> > >
> > > ***“At present, only Figure 2 shows that the proposed method can detect some minority samples, which is not convincing enough for me.”***
> > >
> > > Except for Figure 2, we also conduct an ablation study on two datasets (i.e., GitHub and Instagram) in Table 3 of the paper. From Table 3, we can see that the performance of CMI-GCL without network pruning, i.e., **- pruning** (minus pruning) decreases by comparing it with the performance of our model CMI-GCL, which can demonstrate that network pruning contributes to CMI-GCL in imbalanced datasets.
> > >
> > >
> > > [1] What do compressed deep neural networks forget? arXiv 2019.
> > >
> > > [2] Self-damaging contrastive learning, ICML'21.
> > >
> > > [3] To prune, or not to prune: exploring the efficacy of pruning for model compression, arXiv 2017.

---

### Official Review · Reviewer_zdFn · 2022-07-11

**Rating:** 8
**Confidence:** 4
**Soundness:** 3 good
**Presentation:** 3 good
**Contribution:** 3 good

**Summary:**

In this paper, the authors propose CMI-GCL to generate contrastive pairs in real-world imbalanced graph datasets automatically. At first, it designs the inter-modality GCL module by co-training encoders in different modalities to automatically generate contrastive pairs. To deal with the data imbalance problem, CMI-GCL employs network pruning to prune graph encoders and image encoders to boost the weights of monitor samples during GCL optimization. In addition, it designs an intra-modality GCL module to guarantee that similar nodes in the graph have similar embeddings. Experimental results on two real-world datasets show that CMI-GCL outperforms many state-of-the-art baseline models and shows the strong applicability of the CMI-GCL framework.

**Questions:**

i) In table 3, why does the co-modality module have the biggest contribution to CMI-GCL?

ii) Could you explain more about Figure 2?  why does subset D0.1 can detect the most minority samples in the beginning? Why does subset D init have the lowest percentage in the beginning? Why does D 0.01 have the highest percentage in the end?

iii) In Section 5.5, it claims that the framework CMI-GCL is designed as a play-and-play tool and has very strong applicability to mainstream models. I wonder how many models are already considered in this framework. Could you list some examples except these models listed in Figure 3?

**Limitations:**

Yes.

**Strengths And Weaknesses:**

$\textbf{Strengths:}$

i) The model idea in this paper is novel. Most GCL models focus on benchmark datasets, but they cannot be applied to real-world data. This paper points out the gaps of existing GCL models to real-world data and proposes to co-train encoders across different modalities to generate contrastive pairs. I know there are some cross-modality learning works in image-text, or video-text fields (e.g., CLIP), but co-train the cross-modality encoders in graph-image or graph-text fields is a good try. It looks like that cross-modality GCL has gained excellent performance in two real-world datasets based on the results of the ablation study.  Besides, the idea that applying the network pruning to detect minority nodes in self-supervised learning is also a good try, especially pruning the encoders in different modalities.

ii) This paper is well organized with clear motivations and sufficient comparison experiments in two real-world datasets, which makes the paper easy to understand and follow.

iii) The detailed discussion about experiments is impressive. To evaluate the performance of CMI-GCL, this paper defines two fine-tuning modes for imbalanced datasets and balanced datasets, respectively. By comparing CMI-GCL with state-of-the-art baselines models on two real-world datasets (GitHub and Instagram), CMI-GCL shows its superiority. I think Figure 2 is impressive to show that network pruning is useful to detect minority nodes in GCL pre-training.

iv) As CMI-GCL is developed as a plug-and-play tool, I think CMI-GCL can be considered as a general framework for real-world graph datasets and it can be applied to most real-world graphs with multi-modal data.  Figure 3 describes the performance of the encoder’s combination, showing that CMI-GCL is applicable to mainstream models in different modalities, like BERT and DistilBERT for text content, GCN, GAT, and SAGE for graph, ResNet and Swin Transformer for image content. But I wonder about the number of models that are developed for this framework. With more models considered in CMI-GCL framework, CMI-GCL will be more applicable to real-world datasets.

$\textbf{Weaknesses:}$

i) Some definitions should be cited if it is not defined by this work. For instance, PIEs (pruning identified exemplars) in Section 4.2.2 should be cited as it is from the previous work [1].

ii) More explanations about the experimental results should be provided so that it will facilitate the understanding of model improvement. For instance, in Table 3, why does co-modality have the biggest contribution to CMI-GCL?  Could you explain more about Figure 2?

ii) There are some typos in this paper.
E.g., in section 5.5, CMI-GCL is applicable to --> CMI-GCL is applicable to
Line 240, inspired by GCC we introduce --> inspired by GCC, we introduce
DistillBert --> DistilBert

[1] What do compressed deep neural networks forget? arXiv 2019.

---

> ### Author Response · Authors · 2022-08-02
> **Response to Reviewer zdFn**
>
> Thank you very much for your valuable comments. We will answer your questions and concerns as follows.
>
> Q1: Why does co-modality GCL have the biggest contribution to CMI-CGL based on Table 3?
>
> **A1**: Table 3 lists the ablation experimental results on multiple real-world datasets. We remove the co-modality GCL module to validate the contribution of co-modality GCL to CMI-GCL. We can easily find that the performance of removing co-modality GCL has the biggest drop. The reason behind that is the co-modality GCL includes both the inter-modality GCL module and the intra-modality GCL module. It co-trains pruned encoders in different modalities in the inter-modality GCL module and further co-trains pruned and non-pruned GNN encoders in the intra-modality GCL  module to conduct contrastive learning for learning the more balanced representations.
>
> Q2: Explain more about Figure 2.
>
> **A2**: Figure 2 shows the percentage of PIEs that belongs to the minority classes under different training epochs. From Figure 2, we can find that minority samples in subsets $D_{0.1}$ and $D_{0.01}$ are more likely to be detected via network pruning and the subset $D_{0.1}$ can detect the most minority samples at the beginning, followed by subset $D_{0.01}$. The reason may be that the more imbalanced the data distribution is, it is harder to find minority samples via network pruning at the beginning. However, with the training epochs, CMI-GCL is more powerful to find minority samples even in extremely imbalanced datasets. That's the reason why the subset $D_{0.01}$ has the biggest percentage at the end.
>
> Q3: How many models are included in CMI-GCL?
>
> **A3**: CMI-GCL is designed as a plug-and-play tool to handle most various real-world graph datasets. Most mainstream encoders in text-modal, image-modal, and graph-modal are applicable to our model framework. For instance, in image-modal, CMI-GCL is applicable to CNN-based model (e.g., ResNet, NFNet, ViT,  and ImageNet), transformer-based model (e.g., TNT, Swin Transformer, PiT, and Twins), MLP-based model (e.g., MLP-Mixer, ResMLP, and gMLP). In graph-modal, most gnn models are covered in CMI-GCL (e.g., GCN, GAT, GraphSAGE, and GIN). In text-modal, CMI-GCL is applicable to many popular language models (e.g., BERT, DistilBERT, and DeBERTa).
>
> Q4: Minor issues.
>
> **A4**: Thanks for your suggestion. we have edited these minor issues in the revised paper.

---

> > ### Comment · Reviewer_zdFn · 2022-08-07
> > **New comment after response**
> >
> > Thank you for your response. It addresses my comments. I also see the additional experiments on benchmark data, as pointed out by the other reviewers. Thus, I increase my score to support this paper.

---

> > > ### Author Response · Authors · 2022-08-07
> > > **Reply to score increase**
> > >
> > > Thank you very much for your support and we really appreciate it!

---

### Author Response · Authors · 2022-08-02
**Comment to All Reviewers**

Dear Reviewers:

We really appreciate all of your valuable comments and have made clarifications to all of your questions and concerns in our response. Most reviewers agree that this work is novel and solid. Many questions are related to baselines comparison and experimental details (Q1, Q2 of Reviewer zdFn, Q2 of Reviewer UmCM, Q1, Q2, Q3 of Review D25V), which we have described in the revised paper. In consideration of the limitation of this work we mentioned in this paper and to make our CMI-GCL more convincing, we further apply CMI-GCL to AMiner data, a public paper-citation benchmark graph dataset with raw content. Please refer to the revised appendix file for more details (emphasized in red). You can also find more details about additional experiments on AMiner data in our response. In addition, CMI-GCL aims to be a general GCL model for various real-world imbalanced graph datasets. Hence, CMI-GCL is designed as a plug-and-play model that integrates most mainstream models in different modalities (e.g., image-modal: ResNet, Swin Transformer, PiT, ResMLP. text-modal: BERT, DistilBERT, DeBERTA. graph-modal: GCN, GAT, GraphSAGE, GIN).

We really hope our response can address your concerns and please let us know if you have any more comments.

---

> ### Author Response · Authors · 2022-08-06
> **Discussion Reminder**
>
> Dear reviewers,
>
> Thank you for reviewing our paper. It has been several days since we submitted our response but we have not received your reply.
> We spend a lot of effort in rebuttal and we really hope that you can check whether our responses have addressed your concerns.
> Please let us know if you have any more question and comment. Thank you very much!

---

### Meta-Review · Area_Chair_Cvh9 · 2022-08-26

**Recommendation:** Accept
**Confidence:** Less certain

**Metareview:**

All the reviewers finally give prone-to-accept score 5/5/8 to this paper after the authors' informative communication and adding more experiments on two additional datasets. I agree that the paper in general is novel and the approach is reasonably designed. I suggest to accept this paper while it can be also improved in many aspects especially for clarity as pointed out by the reviewers.

**Award:**

No

---

### Decision · Program_Chairs · 2022-09-14

Accept